# Diversified Techniques for Restructuring Meat Protein-Derived Products and Analogues

**DOI:** 10.3390/foods13121950

**Published:** 2024-06-20

**Authors:** Yuliang Cheng, Yiyun Meng, Shengnan Liu

**Affiliations:** 1State Key Laboratory of Food Science and Resources, School of Food Science and Technology, Jiangnan University, Wuxi 214122, China; 2Collaborative Innovation Center of Food Safety and Quality Control in Jiangsu Province, Jiangnan University, Wuxi 214122, China; 6220111208@stu.jiangnan.edu.cn (Y.M.); 6190111045@stu.jiangnan.edu.cn (S.L.)

**Keywords:** restructuring techniques, restructured meat, meat analogues, texture, protein gel

## Abstract

Accompanied by the rapid growth of the global population and increasing public awareness of protein-rich foods, the market demand for protein-derived products is booming. Utilizing available technologies to make full use of meat by-products, such as scraps, trimmings, etc., to produce restructured meat products and explore emerging proteins to produce meat analogues can be conducive to alleviating the pressure on supply ends of the market. The present review summarizes diversified techniques (such as high-pressure processing, ultrasonic treatment, edible polysaccharides modification, enzymatic restructuring, etc.) that have been involved in restructuring meat protein-derived products as well as preparing meat analogues identified so far and classifying them into three main categories (physical, chemical and enzymatic). The target systems, processing conditions, effects, advantages, etc., of the included techniques, are comprehensively and systemically summarized and discussed, and their existing problems or developing trends are also briefly prospected. It can be concluded that a better quality of restructured products can be obtained by the combination of different restructuring technologies. This review provides a valuable reference both for the research and industrial production of restructured meat protein-derived products and analogues.

## 1. Introduction

The market demand for high-quality protein (mainly from animal meat and plants) in the whole world is expected to grow exponentially over the next 20 years as a consequence of the rapid growth of the global population and increasing public awareness of protein-rich foods [1]. In this case, there happens to be an urgent need to adopt a sustainable approach to meet such requirements. Although the growing demand for meat can be addressed directly by increasing total production, it is necessary for the whole industry to make currently available resources fully utilized. On the one hand, meat by-products like carcass cuts, trimmings, etc., have to be sufficiently used up, for example, to produce restructured meat products, such as sausage and hamburger slices; on the other hand, the search for emerging protein sources, as well as new products of meat analogues, is also a way of helping to alleviate the pressure of increasing demand.

Restructured meat products can be defined as intermediate-value products as these products fetch intermediate value between muscle steak and traditional burgers [2]. The involved processing, known as restructuring, is very useful in the efficient utilization of low-value carcass cuts, trimmings, and meat from spent or culled animals by the application of tumbling, massaging, blade tenderization, and adhesives [3]. People are keen on restructuring meat to produce healthy meat-derived products that are low in fat and cholesterol and could be added with a series of functional ingredients (such as fiber, vitamins, minerals, or antioxidants) [4,5]. Generally, meat restructuring refers to using machinery to extract the matrix protein in the muscle fiber or take advantage of the binding effect of auxiliary additives (such as salt, phosphates, soy protein, starch, carrageenan, enzymes, etc.) to recombine meat scraps. Usually, meat products are sold directly, and a common means to maintain and improve their texture is preheating. Restructuring changes the original structure of meat and rationalizes the distribution of muscle tissue, adipose tissue, and connective tissue. Previous studies have proved that with the support of restructuring techniques, the physical properties of low-grade meat can be improved, and both their texture and sensory attributes can be ameliorated [6].

Meanwhile, it is also necessary for us to explore emerging protein sources and develop meat analogues with a passion to relieve the pressure on the meat industry and promote its sustainability. The physicochemical and structural changes in concentrated protein under thermo-mechanical processing can create a fibrous or layered meat-like texture [7]. Meat analogues refer to a series of products with similar structure, texture, taste, color, and appearance to meat, which are mainly produced by plant protein [8,9] and have the characteristics of high protein and low calories [10]. Thus, the choice of plant protein should be very cautious, considering their functionality, cost, and ease of obtaining [11]. Vegetables, proteins, beans, grains, oils, and fungi are also good resources of high-quality proteins for meat analogues [12]. It is worth noting that meat analogues have characteristics suitable for kosher and halal food markets. Nevertheless, there are still some problems that need to be solved in their production, such as their technical validity in improving texture, the desensitization of soy protein, grains, and other potential ingredients, clean labels, etc.

This review aims to summarize the relevant techniques and methods that have been widely or newly adopted in the processing of restructured meat protein-derived products and analogues. In recent years, based on the continuous pursuit of high-quality and healthy protein foods, restructured meat products, as well as protein-based meat analogues, are becoming more and more favored by the market and its consumers. It is believed that a good knowledge of the relevant processing technologies involved in such products, especially some efficient and novel techniques, is essential for both producers and researchers. In this review, the above-mentioned technologies applied in restructured meat and meat analogues are classified into the following three main categories: physical, chemical, and enzymatic approaches (Figure 1), which are specifically described, discussed, and prospected. Nevertheless, it is worth noting that this review emphasizes physical techniques (e.g., extrusion, heat, high pressure, ultrasonic treatment, pulsed electric field, and the Pi-Vac), chemical additives (e.g., common edible polysaccharides, salt and phosphates, and other functional ingredients) and enzymes involved in the molding and production stages of restructured meat and meat analogues. Other relevant methods like the fabrication of meat analogues using fibrous proteins as raw materials, such as thermo-extrusion [13], high-temperature conical shear cell [14], electrostatic spinning [15], 3D printing technology [16], etc., are not deeply introduced in this review but can be referred from the other two outstanding papers [12,17].

## 2. Physical Restructuring Techniques

### 2.1. Heat Treatment

Heat treatment is the most universal processing technique in restructuring meat products and protein-based meat analogues. The interaction mechanism of heat treatment on meat proteins has been extensively explained. It can be simply interpreted as muscle protein, which aggregates under proper heat treatment, and when cooling, protein molecules tend to cross-link to form an orderly three-dimensional network structure [21]. Fat and water will be trapped in such a network, contributing to flexibility and lubricity. A number of researchers have studied the influence of heating temperature and time on protein gel strength. At high temperatures, the increase in solidification time leads to a decrease in the breaking force, while at low temperatures, the increase in solidification time leads to an increase [22]. Consequently, in the process of heat treatment, temperature and heating time both have significant impacts on the gelling ability, and the latter directly affects the microstructure and palatability of restructured meat products. In addition, plant proteins, such as soy protein, can form high-quality gels after being preheated, so heat treatment is also widely used for restructuring meat analogues [23]. Ohmic cooking has been proven to be an effective form of processing to produce meat analogues by adjusting the cooking temperature and holding time [24]. Super-heated steam (SHS) is a novel drying technique for producing dried restructured meat products [25]. Kim et al. [26] investigated the effect of SHS on the quality characteristics of semi-dried restructured jerky and found that its moisture content, water activity, as well as residual nitrite decreased with increasing SHS temperature and holding time.

With the development of related technologies, heat treatment gradually combined with other means like high-pressure processing and enzyme processing to promote the quality of restructured meat products [27,28]. On this account, some issues, like optimizing the parameters of such integrated processes for preparing restructured meat products or meat analogues, are still worthy of further exploration.

### 2.2. High-Pressure Processing

High-pressure processing (HPP) has been one of the most extensively used non-thermal processing methods in the food industry in recent years [29]. Different from thermal processing, HPP can inactivate microorganisms and enzymes in relatively low temperatures and promote meat texture by gelling proteins, which has a positive effect on the quality of restructured meat products. Figure 2 shows the structure of a high hydrostatic-pressure-generating pump or pressure intensifier, which is the key device of HPP [30]. The current applications and studies of HPP in the meat industry are summarized as follows.

First, for restructured meat, HPP works on many occasions, including muscle enzymes inactivation, meat proteolysis, texture adjustment, pressure-assisted tenderizing process, alterations in muscle ultrastructure or myofibril and the pressure-induced gelation of meat ground, etc. [31]. It has been reported that high pressure has a beneficial effect on meat tenderness [32,33]. Furthermore, the mechanical and textural properties of the heat-induced gels were enhanced by HPP. Cando et al. [27] found that the conformation of the myofibril network treated with HPP (150–500 MPa) after heating was more stable, which could be attributed to the enhancement of -SH groups on the structural stability of myofibrils through non-covalent interactions. Therefore, HHP may lead to partial protein denaturation and thus improve protein functional properties, for instance, the gel-forming ability. So, HPP can assist in heat treatment to induce protein gelation. Velazquez et al. [34] confirmed that better protein networks, higher hardness, and gel strength could be obtained by heat-induced gelation (90 °C) after the high-pressure treatment of 400 to 600 MPa on the flounder paste. These studies point out new inspirations for the development of restructured meat products using HPP, but more extensive explorations are still needed. Moreover, it has been proven that the raw texture of restructured surimi balls can be retained by HPP to a maximum extent, while conventional heat processing usually makes the texture hard and compact [35].

In the aspect of meat analogues, HPP also plays an important role in obtaining an ideal gel mesh structure since meat and meat analogues can be regarded as cross-linked polymer networks [36,37,38,39,40]. The water-holding capacity (WHC) of polymer networks can be explained by the Flory–Rehner theory [36,38], which describes the expansion of cross-linked polymer networks under the action of osmotic pressure (mixing pressure) and elastic pressure [41]. At equilibrium, the totality of these contributions is the pressure applied to the networks. The changes in the microstructure of high-protein products showed that the tighter the protein networks become, the firmer the products are [42]. Therefore, by understanding the interaction between different components of biopolymers, the structural characteristics that play an auxiliary role in the development and improvement of meat analogues can be regulated by some new technology [43]. Predictably, high-pressure technology can be widely applied in the production of restructured meat analogues.

In the meantime, the application of high pressure gradually increases in food processing and storage [30], and existing studies have fully confirmed that this non-thermal technology has an impact on the inactivation of spoilage and pathogenic microorganisms. It follows that HPP can heighten the safety of restructured meat products and extend their shelf life [44]. Research suggests that high-pressure treatment can heighten the quality and shelf life of low-salt naturally cured restructured ham [45,46]. And technology is becoming more and more economically feasible, ensuring solid technical foundations are sustained for restructured meat products. Despite this, HPP usually has a negative effect on the mouthfeel, which has to be minimized by food processors.

### 2.3. Ultrasonic Treatment

Ultrasound facilitates the degradation of myofibrils, which, in turn, bind the meat mixture [47]. Therefore, it can modify the microstructure of recombined meat products by creating micro-cracks in muscle fibers, resulting in significant improvements in taste and sensory properties, such as ameliorated color, increased yield, reduced total fluid release, etc., and without any harmful effect on oxidative stability. Investigations state clearly that ultrasonic treatment has a beneficial influence on improving the sensory acceptance of restructured cooked ham with a 0.75% salt content [48]. The large gel mesh formed by myofibrils helps to retain more water and retard weight loss during storage. Hence, ultrasonic processing shows excellent potential in the preparation of low-sodium, healthy, restructured meat products. Carcel et al. [49] emphasized that ultrasonic technology, due to its advantages, such as low energy consumption and promoting the yield and quality of products, may give impetus to the development of restructured meat products in the industry, opening up another innovative territory.

Ultrasonic technology has also played a big role in producing meat analogues. Up to now, the influence of ultrasound on the properties of modified protein gels has been discussed in various studies. Ultrasonic treatment greatly increases the gel strength, gel yield, rheological properties, WHC, and microstructure of protein gels. The gelation characteristics of whey proteins induced by heating were significantly enhanced after ultrasonic treatment [50]. The high-intensity ultrasonic pretreatment of soybean protein isolate (TSCG) catalyzed by transglutaminase significantly improved its gel strength, WHC, and gel yield, and the obtained microstructure was more uniform and compact [51]. Moreover, ultrasonic treatment could clearly transform the hydrophobicity of the protein surface and ameliorate the microstructure and rheological properties of tofu, which were made from different soybean varieties [52]. Ultrasound technology has immense potential in restructured meat products and is still worthy of in-depth study.

### 2.4. Pulsed Electric Field

Pulsed electric field (PEF), as a novel processing technology, has growing applications in restructured meat products. PEF impinges the cell membrane through electricity, which causes the cell membrane to temporarily or permanently lose semi-permeability, thus improving the dissolution of compounds and nutrients from different substrates [53,54,55,56]. The restructuring process of meat involves the extraction of proteins from membrane-bound muscle fibers, and PEF can increase the release of myofibrils and myosinogen and possibly reconstruct the protein networks [57]. Tenderness is a critical sensory indicator of meat products, and how to improve tenderness has become an important research topic. Arroyo et al. [58] confirmed that the tenderness of longissimus thoracis et lumborum (LTL) muscle samples was improved by pre-PEF treatment (300 and 600 pulses). Nevertheless, studies also showed that high PEF treatment had a negative impact on shear force [59]. It still has great commercial potential to explore the best use of PEF in this area.

### 2.5. The Pi-Vac Elasto-Pack System

The Pi-Vac Elasto-Pack System is a novel technique that wraps hot-boned muscle by inserting it into metal tubes lined with stretchable elastic materials, which could prevent muscle shortening and toughening during rigor [60]. Baugreet et al. [13] illustrated the schematic of the Pi-Vac technique (Figure 3c) reproduced from Meixner and Karnitzschky [61].

It was reported that fresh meat treated by stretching had better cold storage, and its sensory quality was promoted, especially regarding tenderness [62]. The uniformity of muscle shape was improved during cutting [63,64]. Baugreet et al. [13] compared the difference between the Pi-Vac-treated and non-treated beef samples, and it was concluded that Pi-Vac technology could promote the apparent quality and texture of restructured beef steaks (Figure 3a,b). However, there is no ample use of Pi-Vac, and it still has great potential in developing restructured meat products and reducing costs.

Physical processing techniques mentioned above can improve sensory properties, such as cooking loss rate, non-compressed water capacity, water retention, and the cohesiveness of restructured meat and meat analogues, and the microstructure can also be more regular and orderly. They are indispensable techniques for restructured meat and meat analogues. Other means, such as types of microwave processing that form high-quality protein gels [65] and high-moisture extrusion technology that can produce chunked protein-based meat analogues, also have potential applications in restructuring meat and meat analogues. In addition, the extrusion under a low moisture content (<30% moisture) is used for TVP preparation, and the product has an aggregated and more or less expanded conformation. The denatured and aggregated proteins, upon rehydration, give TVP the type of texture resembling meat particles and, therefore, are used as a main ingredient in nonmeat products [12].

## 3. Chemical Restructuring Techniques

Chemical restructuring mainly uses edible polysaccharides, salt, phosphates, etc., to improve the water-holding capacity, elasticity, and gelling ability of meat and meat analogues [12].

### 3.1. Edible Polysaccharides (EPs)

Since studies have demonstrated some harmful effects of excessive meat intake on human health and a lot of researchers are working to prepare restructured meat products either by adding bioactive ingredients (such as dietary fiber, antioxidants, etc.) or replacing harmful ingredients (for example, using plant-derived proteins) [66,67,68]. EPs with both healthy and functional properties are suitable choices for the preparation and improvement of restructured meat products.

Studies found that EPs, as gelling agents, thickeners, emulsifiers, stabilizers, and so on, showed unique functionality in food [69]. Correspondingly, adding dietary fiber could improve the texture properties of restructured products, such as excellent juiciness, by retaining moisture and reducing cooking loss [70]. The different interpretations of EPs derived from cereal in improving the functional characteristics of restructured meat products are summarized in Figure 4 [14]. This review focused on Eps, which are widely used in restructured meat and meat analogues, such as starch, cellulose, hemicellulose, sodium alginate, carrageenan, gum tragacanth, konjac gum, xanthan, etc.

#### 3.1.1. Applications in Restructuring Meat Protein-derived Products and Analogues

Starch. Starch or modified starch plays an important role in the production of restructured meat and meat analogues, acting as adhesives [71], emulsifiers, fat substitutes [72], adhesives, gelling agents and water-retaining agents [73]. It could be mainly attributed to the bi-functional chains that starch has, like the hydrophobic chains that form a network structure with fat molecules through the hydrophobic interaction, while the hydrophilic one forms a network with water molecules so as to achieve excellent textural, rheological, and sensory properties. Carballo et al. [74] found that starch had a positive effect on reducing cooking loss and increasing the hardness, chewiness, and penetration force of bologna sausage.

It was found that, compared with native starch, modified starch had a better performance on restructuring meat. Sausages made from natural WX cornstarch have large and uneven pores, while those made from OSA starch possess relatively compact ones [73]. OSA-modified corn starch has great potential to improve the texture of restructured meat products.

In addition, the combination of starch with other ingredients (such as carrageenan, soy protein isolates, and milk protein) is also conducive to obtaining better physical texture and sensory properties. In America, native starch and its derivatives (unmodified and modified forms), as important substances to improve texture, storage, and processing, have been officially approved and widely used in restructuring meat and meat analogues and are regulated by the Food Safety and Inspection Service of the United States Department of Agriculture [14]. Since the type of starch that works depends on what the final products require, various assessments and tests need to be carried out for the preparation of commercially restructured products [75].

Cellulose and Hemicellulose. At present, two main types of cellulose, including carboxymethyl cellulose (CMC) and microcrystalline cellulose (MCC), have been used for restructuring meat and meat analogues as water-retaining agents, adhesives, and stabilizers and exhibited positive effects on the physiochemical and sensory properties of restructured products [76,77,78]. It is worth mentioning that, due to its large surface area, high strength, high stability, excellent rheology, and emulsifying capacity, nanocellulose has been proven to be an effective alternative for restructuring meat and meat analogues. Researchers compared different samples with cellulose nanofibers (CNFs) either added or not, and found that, at the same fat level, samples added with CNF had a denser network, smaller cavities, and the structure became more continuous and compact, resulting in a better texture and reduced cooking loss [79]. It can be concluded that the rigid network formed by adding CNF is cross-linked with the protein matrix, which helps to obtain more fat droplets to form a more homogeneous structure. And beyond that, the application of cellulose as a fat analogue in restructured products is introduced in a later section.

Hemicellulose mainly exists in the form of β-glucan or arabinoxylan and plays filling, bonding, and thickening roles in restructuring meat and meat analogues. For example, adding β-glucan and inulin to hamburger patties can achieve better texture, water retention, cooking yield, and overall acceptance. β-glucan made fat and meat tightly connected and formed a dense matrix that can hold water [80]. Adding 0.15% and 0.30% arabinoxylan ferulic acid to Frankfurt sausage can improve water-holding capacity, hardness, grain diameter, titratable acidity and the sheer force of sausage samples and enhance antioxidant capacity [81].

Compared to cellulose, hemicellulose has attracted more interest from researchers because of its general health-related potential. β-glucan has been a hot topic for more than a decade as an immune stimulator and healthy factor like some cholesterol-lowering drugs, antioxidants, etc. [82]. And arabinoxylan has also been proven to have significant health benefits as an anti-inflammatory and anti-cancer agent. Based on this, researchers found that the application of hemicellulose in foods helped to obtain unique nutritional and physiological properties. For example, studies found that the intervention of arabinoxylan could help to delay lipid peroxidation, and subsequent oxidative deterioration [83]. It can be predicted that hemicelluloses such as β-glucan have great potential in high-end restructured meat products and meat analogues in an environment where health is increasingly valued by consumers.

Sodium Alginate (SA), Carrageenan and Gum Tragacanth (GT). In food matrixes where proteins and indigestible polysaccharides, such as sodium alginate, carrageenan, GT, etc., coexist, the stability and rheological properties of the system can be affected by the interaction between two biopolymers. This can be divided into two cases. In food with a low polysaccharide content and weak net attraction, polymer bridges between emulsion droplets may be formed to cause flocculation. On the other hand, since the surface of protein-stabilized droplets is coated with enough interacting polysaccharides, the emulsion may not be flocculated due to the re-stabilization of the secondary spatially stabilized polysaccharide layer. SA is one of the most commonly used indigestible polysaccharides in restructuring meat and meat analogues. Choosing SA as a gelling agent can achieve a lot of advantages, especially its highly regulable ionotropic gelation processes and relatively low costs [84,85]. Nykyforov et al. [86] found that restructured fish products obtained the maximum value of sensory evaluation with a 2-2.5% addition of SA plus a 5% concentration of calcium chloride in the solution. It also reported that adding no less than 3% SA could achieve sufficient gelation of minced meat [87]. Kim et al. [88] investigated the effect of SA combined with two common-used hydrocolloids (konjac and carrageenan) on the quality of restructured duck ham and confirmed that adding 1% SA or 0.5% SA plus 0.5% konjac could achieve the best quality.

Carrageenan, as a typical anionic polysaccharide, can induce bridging flocculation [89]. Verbeken et al. [90] observed that the restructured meat products added carrageenan and, indeed, formed a three-dimensional gel network via confocal microscopy, which is consistent with the results of rheological analysis and the presence of carrageenan in the discrete region, suggesting the possibility of an interaction network between carrageenan and the protein. Based on the above understanding, researchers have studied either using carrageenan alone or in combination with other ingredients in a variety of restructured meat and plant-based meat analogues and made it clear that carrageenan and protein could produce a gelatinization effect, which is beneficial to form a complete structure with the advantages of retaining water, improving flavor, tenderness, and stability [6,91,92,93,94].

As a tasteless and edible natural polysaccharide, GT has a similar working path to carrageenan, and it also has great potential in the production of restructured meat and meat analogues. The anionic groups of this negatively charged polysaccharide can be electrostatically attached to the acidic residues of the protein to form a solid and complex interface layer that completely covers oil droplets; thereby, GT can stabilize protein emulsions at an acidic pH or high ionic strength. In the GT/protein system, GT connects to oppositely charged proteins via electrostatic interactions, which can lead to the formation of soluble complexes or aggregation depending on the pH, ionic strength, protein/GT ratio, and total biopolymer substance concentration [95]. The mixture of GT and proteins undergoes a series of apparent changes at critical pHs [96]. Studies have revealed that adding GT to the lengthened and restructured lamb chops could improve the cooking yield, binding degree, texture, and overall acceptance [97]. Therefore, tragacanth has great potential in both restructuring meat products and meat analogues.

In summary, the reason why polysaccharides can work in restructuring meat and meat analogues is mainly attributed to the following specific aspects. On the one hand, hydrogen bonds, hydrophobic interactions, and spatial interactions are major forces in the formation of the hydrocolloid–protein gel network [98]. On the other hand, the sensory nature of restructured products can be changed due to the formation of a soluble polysaccharide–protein complex above the isoelectric point [99]. In addition, sulfate in polysaccharides may interact with those positively charged sites on the protein (such as -amino, guanidine, and imidazole) and affect the microstructure. The effect depends on the number and distribution of these sites as well as the net charge of the protein. However, there are still a variety of polysaccharides that have been widely used for preparing restructured meat and meat analogues, such as konjac gum [100,101], xanthan [102,103], low methoxyl pectin [104,105], gellan [106], galactomannan [107], etc., giving restructured products better textural and sensory properties.

#### 3.1.2. Applications in Low-Fat-Restructured Products

Excessive fat intake has been proven to be one of the most important factors leading to many diseases. As healthy foods become a market trend, consumers are increasingly favoring low-fat foods. However, fat is of great significance for meat products, providing flavor, nutrition, texture, and sensory characteristics. How to reduce total fat without compromising the texture, sensory, and overall acceptance of meat products has become a hot topic for researchers. It is a promising strategy to use polysaccharides to compensate for fat loss.

Starch, with a similar particle size to that of a fat droplet, showed excellent performance in the production of low-fat meat products [108,109]. Therefore, meat products can be reformulated by adding starch or its derivatives (modified forms) as fat substitutes. Jairath et al. [110] found that cornstarch acting as a fat substitute could effectively improve the water-holding capacity and sensory properties of restructured veal sausages. In similar cases, the moisture in restructured meat and meat analogues is retained, and shortcomings such as crispiness are avoided [111]. However, it is also reported that the addition of corn starch has a negative effect on the cooking loss of restructured meat [112].

In addition to being a good adhesive for restructured meat, as mentioned above, cellulose also plays an important role in the production of low-fat-restructured products. Studies showed that both CMC and MCC fibers had positive effects. However, it is worth noting that, compared to the destructive effect on protein networks by adding CMC, MCC showed no disruption [78]. It can be concluded that MCC is more conducive to maintaining the integrity of the protein gel network, and CMC may reduce the stiffness of reduced-fat sausages. Their effects mainly depend on the molecular weight and degree of substitution. Studies showed that a decrease in molecular weight led to a decrease in stability [113]. It is worth mentioning that due to its excellent water-holding capacity, amorphous cellulose can increase the viscosity of restructured products, thereby providing similar juiciness and textural properties as fat. Studies found that the physical, chemical, and sensory properties of low-fat Bologna sausages were greatly improved since amorphous cellulose was added. This combination of amorphous cellulose and pork rind also provided new ideas for the preparation of low-fat sausages [114].

Hemicellulose, like β-glucan, has also been widely used as a fat substitute in meat products. Studies found that low-fat meatballs with β-glucan added had better hardness, chewiness, elasticity, cohesion, and a smoother surface than high-fat ones [115]. And it indicated that β-glucan enhanced water retention by blocking the formation of protein networks. The addition of β-glucan as a fat substitute improves the uniformity of restructured meatballs. Moreover, β-glucans derived from yeast and mushrooms have also been added to meat analogues as fat substitutes [116].

Furthermore, researchers have found that when konjac gum is used in combination with other ingredients (such as starch, gellan gum, and carrageenan), it can also be used as a fat analogue in low-fat-restructured meat products. Chin et al. [117] compared the three-dimensional structure of low-fat Bologna samples with different mixtures of EPs added either with or without soy protein at a 2000× magnification, and it showed that the protein matrix of low-fat Bologna batter prepared with konjac flour plus starch had large pores, while the protein paste prepared with konjac flour plus carrageenan and starch seemed relatively compact. To date, konjac gum has been widely used for preparing low-fat frankfurter sausages [100,101,118,119], Bologna [117], fresh sausages [120], and pork nuggets [121]. In addition, Gadekar et al. [122] evaluated the effects of inulin, chitosan, and carrageenan on the overall quality of low-fat-restructured goat meat products, respectively, and confirmed that the above three functional ingredients could be used to improve physicochemical, sensory and textural attributes.

### 3.2. Salt and Phosphates

Adding salt into minced meat helps to extract myofibrillar proteins, which contribute to the enhancement of binding and phosphates, strengthen the effect of salt and play a big role in restructured meat and meat analogues to improve water-holding capacity and resist shrinkage [123,124,125]. Commonly used phosphates are pyrophosphate, tripolyphosphate, hessian phosphate, sodium phosphate, etc. [126]. Since the role of salt and phosphates in processed meat as well as restructured meat has been extensively delved into and the underlying mechanism is well explained, they are no longer a hot topic in recent years. On the contrary, studies on the replacement of salt and phosphates have attracted considerable interest.

Previous studies have proven that small amounts of salt and phosphates can promote the overall quality of restructured meat products [126,127,128,129]. Low salt content could lead to an increased cooking loss rate (6.27 vs. 3.25%), moisture, and C* value but declined hardness [127]. Accordingly, salt content mainly affects the hardness, water activity, expressible water value, chewiness, and slicing ability of target products [128].

It is worth mentioning that excessive sodium intake has been proven to be one of the most important factors leading to a high risk of hypertension, stroke, and premature death from cardiovascular disease. In most industrialized countries, however, sodium intake often exceeds that of local nutritional recommendations. Some studies aimed to reduce the sodium content in restructured meat products by partially replacing NaCl with KCl, CaCl_2_, MgCl_2_, etc., to maintain the required ionic strength [112,130,131]. Horita et al. [132] compared a set of reduced sodium Frankfurter sausages treated with a mixture of three kinds of hydrochlorides (NaCl, KCl, and CaCl_2_) and found that samples with an ionic strength equivalent to 2% NaCl had better sensory properties. It was confirmed by SEM analysis that in samples with a higher sodium chloride content and ionic strength, the higher extraction rate of myofibrillar protein resulted in a porous, uniform, and dense protein matrix with excellent cohesion between the continuous and dispersed phases. Consequently, it is proposed to be a reasonable and efficient solution to reduce sodium content by partially replacing sodium chloride with other common hydrochlorides.

Meanwhile, excessive phosphate intake has also been proven to be an inducing factor of imbalance in calcium, magnesium, and iron in the human body and can probably lead to bone disease [133]. Therefore, it is strongly recommended to reduce the use of phosphates in the meat industry [134]. Accordingly, researchers carried out experimental studies on the replacement of phosphates. Schutte et al. [135] found that iota carrageenan (iota-CGN) could be substituted for sodium tripolyphosphate (STPP) (up to 0.35% STPP and 0.2% iota-CGN) to produce reduced STPP ham. Öztürk-Kerimoğlu and Serdaroğlu [136] replaced STPP with inulin and sodium in restructured chicken steaks and concluded that the incorporation of inulin and sodium carbonate could compensate for the beneficial effects of STPP.

At low ionic strength, the unfolding of protein molecules is inhibited by the buffering effect, which can widen the isoelectric point of plant proteins and, thus, slow down their swelling. However, pH is still the decisive factor affecting the degree of swelling. A previous study of great value confirmed that the pH and ionic strength of marinades can be effective tools to enhance the water retention of model meat analogues [137]. Figure 5 shows the swelling level of meat analogues in baths with different levels of pH and ionic strength, which helps to develop salt-reduced meat analogues. In addition, the water-holding capacity can also be improved by changing the crosslinking density without salt added. With these tools, food manufacturers can develop meat analogues that are more like real meat and increase the final product yields as well.

### 3.3. Other Substances

Other substances, such as soy protein [66,124], barley flour [138], citric acid [139], egg white [140], sodium nitrite [141], etc., can also improve the quality of restructured meat and its analogues and reduce the total cost. According to existing studies, citric acid can influence the content of myofibrillar protein as well as its secondary structure [139]. According to SEM results, the protein networks of 0.2% citric acid-treated restructured fish products (RPCs) and surimi products (SPCs) became more refined, thus improving the Aw, dry matter, color, texture, and acceptability of restructured meat products.

Moreover, it is worth mentioning that the addition of emulsion gels can solve problems that result from directly adding liquid oil to restructured meat products. In particular, inulin-based water-in-oil emulsion hydrogels, as a complex colloidal matrix with the coexistence of emulsion and gel structures, have stable rheological properties and can better stabilize oil [142]. Meanwhile, the application of polysaccharides in hydrogel preparation still has great potential [143,144,145]. Emulsion hydrogels have a good prospect in the production of restructured meat and meat analogues due to their simple technical requirement, low cost, and similar properties to animal fats [146]. In addition, glycosylation is one of the potential methods to improve the properties of protein gel [147]. Glycosylation is considered to improve the stability of the emulsion and has a good application prospect in the production of restructured meat and meat analogues.

## 4. Enzymatic Processing

At present, consumers are increasingly demanding to use less or even no additives to achieve clean labels. The application of enzymes in restructured meat and meat analogues will provide advantages for consumers and the meat industry.

### 4.1. Enzymatic Restructuring Techniques

Enzymatic restructuring techniques mainly use enzymes to catalyze the polymerization and covalent, cross-linking reaction of myofibrillar proteins and other homologous or heterologous proteins, such as casein, soy protein isolates, etc., resulting in enhanced gel networks and gel stability, so as to restructure the minced meat under certain conditions. Transglutaminase (TGase) is the most extensively used enzyme in restructured meat products and meat analogues [110,148,149,150,151,152,153,154,155,156].

TGase can catalyze the acyl-transfer reactions of free amino acids such as lysine and glutamine residues and γ-carboxyl amide groups for protein cross-links [157]. Figure 6 shows the schematic illustration of the cross-linking reaction between the Gln and Lys residues of proteins catalyzed by TGase [15]. Since TGase has the ability to enhance protein gel networks, its application in restructured meat and meat analogues has attracted a lot of attention. Differing from animal TGase, microbial-derived TGase (MTGase) has a low molecular weight, a simple preparation process, high yield, and easy biodegradation. Hence, MTGase has become the first choice for enzymatic restructuring [154]. It has been fully confirmed that MTGase can help form ε-(γ-glutamyl)-lysyl crosslinks, which are relatively stable and not easy to break and, thus, can strengthen protein networks. According to previous studies, MTGase processing promoted the cohesion, texture, and sensory properties of restructured meat products and meat analogues [151,153,154]. The SEM results showed that the gel network of the control sample without TGase was looser than that of the restructured ones treated with TGase. In addition, the sample with 1.0% TGase formed larger and more complete gel clusters [156].

At present, existing research on the effects of TGase processing mainly focuses on the improvement of the physiochemical properties of restructured meat or fish products. However, it remains to be further studied in terms of plant-based meat analogues. Predictably, the effect of TGase on denatured protein is quite different from that of natural protein. Studies showed that TGase could greatly improve textural parameters and decrease oil absorption as well as the expressible moisture of soy-based meat analogues [149]. The combination of TGase and sodium alginate can effectively improve the texture of protein-based meat analogues [152,155]. The protein networks of samples treated with TGase are relatively tighter, indicating a positive effect on the texture of restructured meat products [155].

Additionally, TGase has been applied in preparing low-fat-restructured products. The combination of TGase and carrageenan caused a significant color change in restructured fish-cooked ham [158]. Sodium caseinate treated with MTGase could take the place of extracted animal fat, and overall sensory properties of the restructured product showed no significant difference from conventional processed meat [154]. Although TGase has been widely used for restructuring meat products, the optimization of TGase processing conditions such as processing time, temperature, dosage, and other parameters depending on different raw materials is necessary.

### 4.2. Applications in Flavor Improvement

How to simulate the flavor of real meat has become one of the biggest challenges in preparing restructured meat and meat analogues, especially for emerging plant-based meat. Researchers have made attempts to use enzymatic hydrolysates as flavor substances in such products. Previous studies have revealed that the main aromatic substances of real meat include several sulfur-containing and nitrogen-containing compounds formed by amino acids and sugars at high temperatures and trace amounts of aldehydes, ketones, alcohols, and furans [159,160]. Researchers used the Maillard reaction that occurred between enzymatic hydrolysates of animal or plant proteins and reducing sugars to produce specific aromatic substances to compensate for the flavor defects of meat analogues [161,162]. In particular, Lotfy et al. [161] compared the effect of conventional (CBF/E-CBF) and microwave (MBF/E-MBF) heating on the preparation of encapsulated beef-like flavorings from enzymatically hydrolyzed mushroom proteins. As shown in Figure 7, sample MBF shows significantly higher scores of beef-like and sulfurous notes than sample CBF, and it indicates that beef-like flavorings may be generated using microwave heating.

The development of restructured meat and meat analogues is still in its early stage. Especially for the latter, only a better meat substitute can obtain higher market acceptance. In addition to the aspects mentioned above, some other possible applications of enzymatic processing, such as the potential of protease in allergens, odor component removal, etc., need to be further explored.

## 5. Conclusions

Diversified techniques for restructuring meat protein-derived products and analogues can be generally classified into three categories: physical, chemical, and enzymatic. Among them, physical restructuring technologies (e.g., extrusion, heat, HPP, ultrasound, PEF, the Pi-Vac, etc.) are the most commonly used, which have clear effects on the improvement of overall quality and are very beneficial to clean labels. However, the choice of methods and on-site technical parameters are too dependent on different raw materials and have adverse effects on mouthfeel (e.g., HPP), shear force (e.g., PEF), and high cost (e.g., the Pi-Vac). Chemical restructuring can improve the quality of restructured meat and meat analogues by adding functional ingredients, such as edible polysaccharides, salt, phosphates, etc., as well as glycosylation modification. Edible polysaccharides not only improve the apparent texture but also reflect unique functionality, and they also have good performance in preparing low-fat-restructured foods and fat substitutes. It has been extensively proven that the addition of salt and phosphates can improve the quality of restructured meat products and meat analogues. But, considering their adverse effects on health, it has become a trend to reduce the use of salt and phosphates or even not to use them. It is worth noting that emulsion gels also show significant advantages and prospects in replacing fat. However, the chemical restructuring approach is not conducive to the realization of clean labels. In contrast, enzymatic processing is an efficient and clean restructuring technology that has attracted the attention of most researchers. However, the optimization of enzymatic processing conditions, as well as the exploration and application of multifunctional enzymes (e.g., for desensitization, odor elimination, etc.), still need to be further investigated. In general, researchers are more inclined to study the integration of different restructuring approaches, such as heat treatment combined with high pressure or enzymatic processing or edible polysaccharides combined with TG enzymes for the preparation of restructured meat protein-derived products and analogues. Better quality can be obtained by combining different types of restructuring techniques. This work provides a valuable reference both for research and the industrial production of restructured meat protein-derived products and analogues.

## Figures and Tables

**Figure 1 foods-13-01950-f001:**
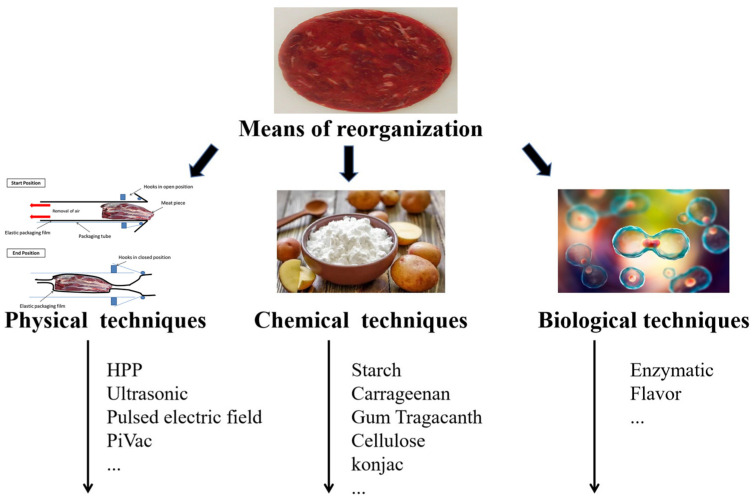
Various techniques for restructuring meat and meat analogues (classified into three categories: physical, chemical and enzymatic). The images are referenced from various sources. (reproduced from [12], with Elsevier.com, 2020; reproduced from [18], with Institute of Food Science & Technology.com, 2017; reproduced from [19,20], with Elsevier.com, 2019 and 2021).

**Figure 2 foods-13-01950-f002:**
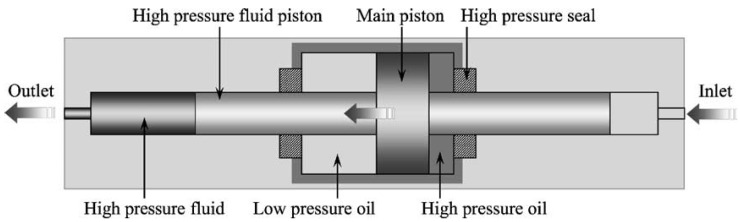
The structure of high hydrostatic pressure pump or pressure intensifier (adapted from [30], with Elsevier.com, 2005).

**Figure 3 foods-13-01950-f003:**
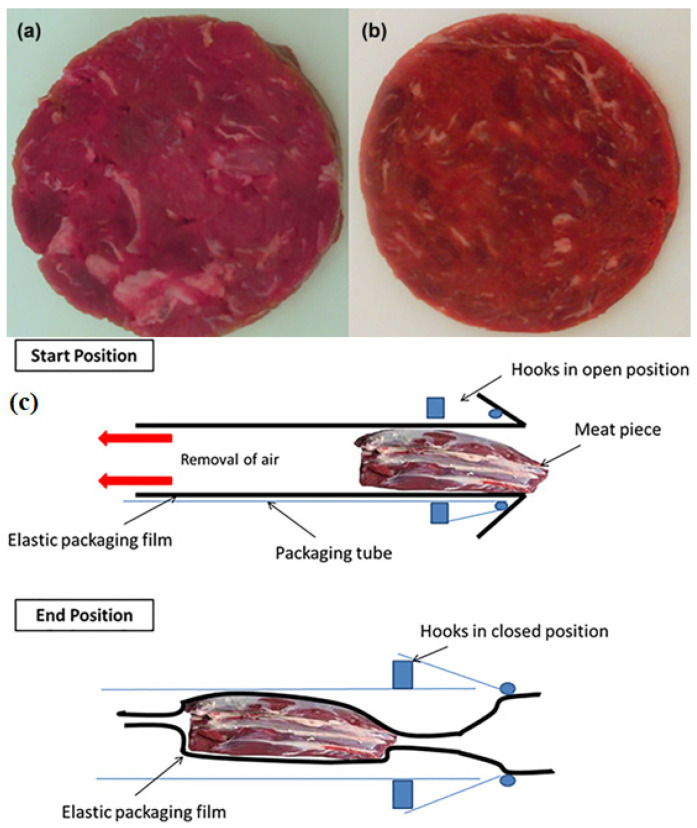
Restructured beef steaks without (**a**) and with (**b**) the Pi-Vac application, and (**c**) schematic diagram of the Pi-Vac system (adapted from [13], with Institute of Food Science & Technology.com, 2017).

**Figure 4 foods-13-01950-f004:**
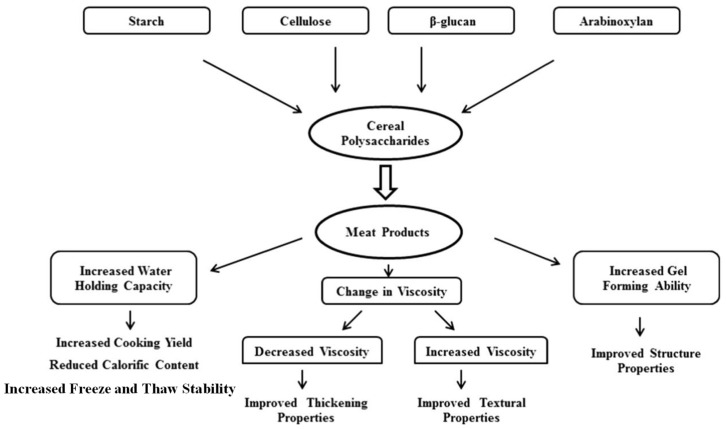
Different interpretations of cereal EPs on improving the functionalities of restructured products.

**Figure 5 foods-13-01950-f005:**
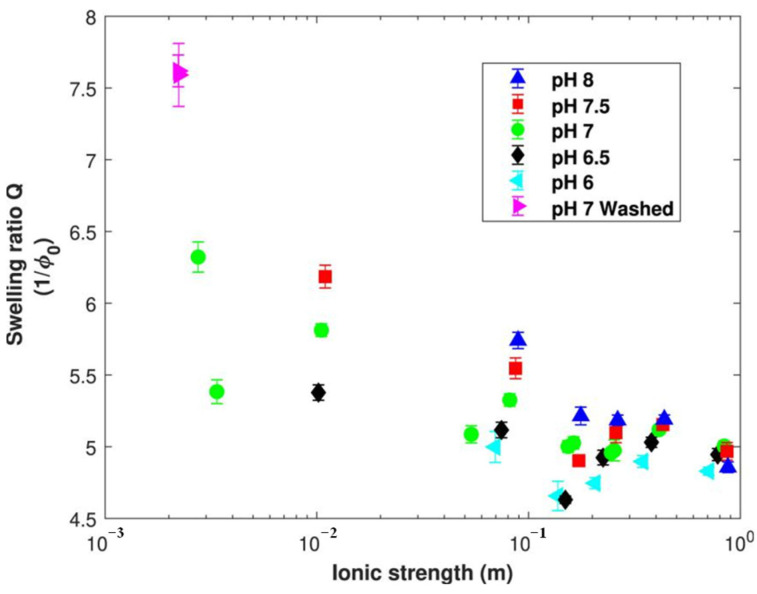
Maximum level of swelling expressed as a function of bath ionic strength and pH. The bath: a phosphate-buffered saline solution or water in the case of the lowest ionic strength. Error bars are the 95% confidence intervals with n = 3 [137].

**Figure 6 foods-13-01950-f006:**
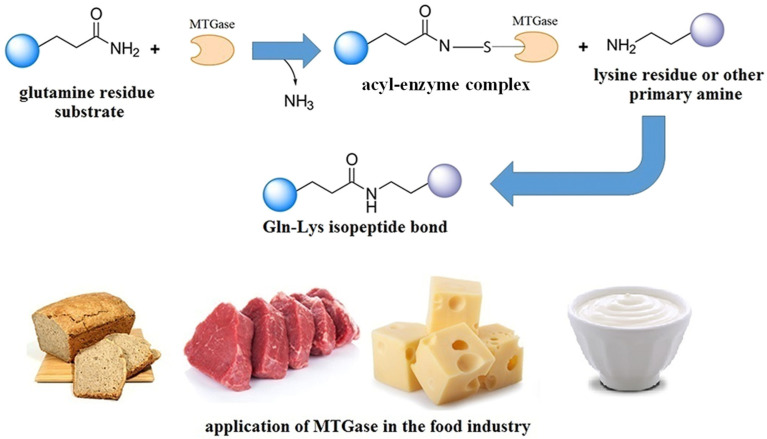
Schematic illustration of the cross-linking reaction between Gln and Lys residues of proteins catalyzed by TGase (reproduced from [15], with Elsevier.com, 2021).

**Figure 7 foods-13-01950-f007:**
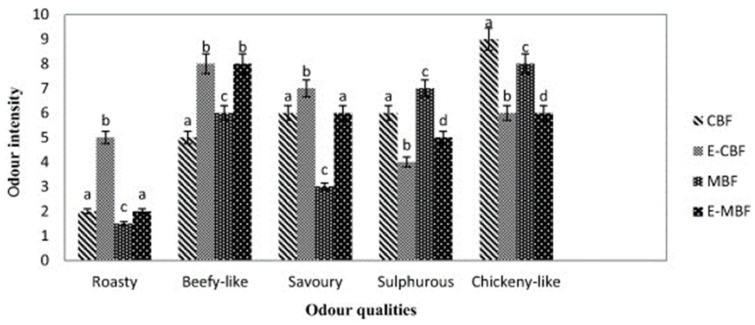
Sensory profile of different flavoring samples. Odor intensities followed by different superscript letters for each odor quality are significantly different (*p* < 0.05) (adapted from [161], with Elsevier.com, 2015).

## Data Availability

No new data were created or analyzed in this study. Data sharing is not applicable to this article.

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
