# Peer review of "Diversified Techniques for Restructuring Meat Protein-Derived Products and Analogues"

_foods, 2024, doi:10.3390/foods13121950_

Round 1

Reviewer 1 Report

Comments and Suggestions for Authors

The review article titled “Diversified techniques for restructuring meat protein-derived products and analogues” provides a good review of this topic. However, I think some minor observations should be considered.

The manuscript is interesting and I think this type of information is worth publishing. However, I think some changes should be made to the manuscript, I have attached a document with observations and suggestions.

Comments on the Quality of English Language

I'm not a native, but I think English needs editing from a native speaker.

Reviewer 2 Report

Comments and Suggestions for Authors

Dear Authors,

The manuscript requires significant revisions to improve clarity, incorporate novel techniques, and provide a structured and differentiated analysis of real meat and meat analogues. The suggestion is to concentrate on meat analogues alone.

Comments are attached.

Comments on the Quality of English Language

In general, in the whole manuscript, the authors need to simplify some of the sentences and improve their structure for better flow and coherence. Moderate English editing is required.

Round 2

Reviewer 2 Report

Comments and Suggestions for Authors

Dear Authors,

By narrowing the scope exclusively to meat analogues production, you can enhance the quality and impact of your review. This distinction is crucial to prevent confusion and enrich the novelty of the review. The rationale behind this suggestion is the lack of clarity regarding the term "reconstructed meat." Your references to burgers and sausages (Line 36), which are distinct product categories, do not align with the concept of reconstructed products. Furthermore, reconstructed meat products belong to a relatively older category that does not necessitate a contemporary explanation, particularly considering the extensive literature on additives like polyphosphates, starch, cellulose, carrageenan, and other hydrocolloids commonly used in such products.

By focusing solely on meat analogues, the authors have the opportunity for a deeper exploration of the latest advancements and emerging trends in this evolving field. This will make the review more relevant and timely, and allow for a more focused and streamlined discussion.

Comments on the Quality of English Language

Moderate English editing is required.
